# Exploring the Outcome of Disappearance or Small Remnants of Colorectal Liver Metastases during First-Line Chemotherapy on Hepatobiliary Contrast-Enhanced and Diffusion-Weighted MR Imaging

**DOI:** 10.3390/cancers15082200

**Published:** 2023-04-08

**Authors:** Piero Boraschi, Roberto Moretto, Francescamaria Donati, Beatrice Borelli, Giuseppe Mercogliano, Luigi Giugliano, Alessandra Boccaccino, Maria Clotilde Della Pina, Piero Colombatto, Stefano Signori, Gianluca Masi, Chiara Cremolini, Lucio Urbani

**Affiliations:** 1Department of Diagnostic and Interventional Radiology, and Nuclear Medicine, Azienda Ospedaliero-Universitaria Pisana, Via Paradisa 2, 56124 Pisa, Italy; f.donati@med.unipi.it (F.D.); c.dellapina@ao-pisa.tocana.it (M.C.D.P.); 2Unit of Medical Oncology 2, Azienda Ospedaliero-Universitaria Pisana, Via Roma 67, 56126 Pisa, Italy; roberto.moretto@ao-pisa.toscana.it (R.M.); beatrice.borelli@phd.unipi.it (B.B.); a.boccaccino@studenti.unipi.it (A.B.); gianluca.masi@unipi.it (G.M.); chiara.cremolini@unipi.it (C.C.); 3Department of Translational Research and New Technologies in Medicine and Surgery, University of Pisa, Via Risorgimento 36, 56126 Pisa, Italy; 4Department of Radiology, University of Naples “Federico II”, Via Pansini 5, 80131 Naples, Italy; giuseppe.mercogliano@aslcaserta.it (G.M.); luigi.giugliano@aslnapoli2nord.it (L.G.); 5Hepatology Unit, Azienda Ospedaliero-Universitaria Pisana, Via Paradisa 2, 56124 Pisa, Italy; p.colombatto@ao-pisa.toscana.it; 6General Surgery, Azienda Ospedaliero-Universitaria Pisana, Via Paradisa 2, 56124 Pisa, Italy; s.signori@ao-pisa.toscana.it (S.S.); l.urbani@ao-pisa.toscana.it (L.U.)

**Keywords:** colorectal liver metastasis, disappearing metastases, residual tiny metastases, hepatobiliary contrast-enhanced magnetic resonance imaging, diffusion-weighted imaging

## Abstract

**Simple Summary:**

Although the extensive use of highly active oncological treatments has led to an increased rate of secondary resections in patients with colorectal liver metastases, it has led to questions regarding the correct surgical management of disappearing liver metastases (DLM) and small residual lesions during chemotherapy. Our study aimed to evaluate the clinical outcome of DLM and small remnant liver metastases (≤10 mm) assessed by means of hepatobiliary contrast-enhanced and diffusion-weighted MR imaging (DW-MRI) in metastatic colorectal cancer patients with first-line chemotherapy treatment in order to support the clinical management and decision-making process for these liver lesions. Our results showed that DLM assessed via hepatobiliary contrast-enhanced and DW-MRI very probably indicate a complete response in patients without chemotherapy-induced sinusoidal obstruction syndrome. In these patients, a follow-up with liver MRI can be considered, and resection should be performed in the case of disease relapse. The surgical removal of small remnants of liver metastases should always be advocated whenever technically possible.

**Abstract:**

We aimed to evaluate the outcome of the disappearance or small remnants of colorectal liver metastases during first-line chemotherapy assessed by hepatobiliary contrast-enhanced and diffusion-weighted MR imaging (DW-MRI). Consecutive patients with at least one disappearing liver metastasis (DLM) or small residual liver metastases (≤10 mm) assessed by hepatobiliary contrast-enhanced and DW-MRI during first-line chemotherapy were included. Liver lesions were categorized into three groups: DLM; residual tiny liver metastases (RTLM) when ≤5 mm; small residual liver metastases (SRLM) when >5mm and ≤10 mm. The outcome of resected liver metastases was assessed in terms of pathological response, whereas lesions left in situ were evaluated in terms of local relapse or progression. Fifty-two outpatients with 265 liver lesions were radiologically reviewed; 185 metastases fulfilled the inclusion criteria: 40 DLM, 82 RTLM and 60 SRLM. We observed a pCR rate of 75% (3/4) in resected DLM and 33% (12/36) of local relapse for DLM left in situ. We observed a risk of relapse of 29% and 57% for RTLM and SRLM left in situ, respectively, and a pCR rate of about 40% overall for resected lesions. DLM assessed via hepatobiliary contrast-enhanced and DW-MRI very probably indicates a complete response. The surgical removal of small remnants of liver metastases should always be advocated whenever technically possible.

## 1. Introduction

Colorectal cancer (CCR) is the third-most-common cancer in both sexes and ranks second in terms of mortality worldwide [1]. In most cases, it develops from adenomatous polyps, benign cell proliferations visible through colonoscopy, which can develop into malignancy in about 10-15 years [2,3]. CCR can remain asymptomatic for a long time, so screening programs for early diagnosis and treatment are essential [4,5]. Nevertheless, from the available data, the percentage of CCR with synchronous liver metastasis is 15%, and the 5-year cumulative risk of metachronous metastasis is 16%. Taking into account the incidence and prevalence of CCR, it can be estimated that the number of cancers with synchronous liver metastases is 5400 per year, and the number of metachronous liver metastases is 9000 per year [6]. Only 10–15% of patients will be candidates for surgical resection at the time of diagnosis [7,8]. Therefore, chemotherapy plays a key role in the treatment of CCR. It is used in the treatment of both unresectable colorectal liver metastases (CRLM) and surgically resectable patients as neoadjuvant therapy [9]. Advances in chemotherapy have brought a reduction in size, not only in primary colorectal cancer but also for liver metastases to the point of their radiological disappearance [10]. However, radiological disappearance of CRLM does not equal a complete pathological response (pCR).

Despite recent improvements in systemic treatment for metastatic colorectal cancer (mCRC) patients, resection of metastases remains the unique chance to cure or to prolong disease-free remission in selected patients with CRLM. The availability of active systemic therapies combined with growing advances in surgical techniques and the widespread multidisciplinary approach have broadened the group of patients deemed eligible for hepatic surgery [11,12]. Whilst the extensive use of more recent and highly active oncological treatments has led to an increased rate of secondary resections, it has questioned the correct surgical management of disappearing liver metastases (DLM) and small residual lesions during chemotherapy. Indeed, as reported by several retrospective series, radiological complete response assessed by computed tomography (CT) imaging corresponds to a real complete response in less than 20% of cases [13,14,15].

The first imaging modality used to detect and characterize CRLM and evaluate resectability is usually contrast-enhanced computed tomography (CT). CT remains the workhorse and is the only useful tool in case of diffuse metastatic liver disease and/or metastatic extra-hepatic disease. In the last few years, liver magnetic resonance imaging (MRI) with hepato-specific contrast agents, such as gadoxetic acid and diffusion-weighted imaging, has been introduced for the management of patients with CRLM eligible for secondary resection [16]. Notably, liver MRI has demonstrated higher sensitivity for small lesions (<10 mm) and diagnostic accuracy in terms of morphological characterization, in particular, after neoadjuvant chemotherapy and in the case of fatty liver with respect to CT scan [5,10,17]. Therefore, MRI is considered the most cost-effective strategy for detecting liver metastases eligible for surgical resection and should be seen as the gold-standard technique.

Drawing on these considerations, we evaluated the clinical evolution of DLM and small residual liver metastases (≤10 mm) assessed by means of hepatobiliary contrast-enhanced and diffusion-weighted MR imaging (DW-MRI) in metastatic CRC patients with first-line chemotherapy treatment in order to support the clinical management and decision-making process for these liver lesions.

## 2. Materials and Methods

### 2.1. Patients’ Population

Consecutive patients with CRLM referred to Pisa University Hospital from November 2016 to December 2020 and with at least one DLM or residual liver metastases less than or equal to 10 mm, assessed via hepatobiliary contrast-enhanced and diffusion-weighted MR imaging during first-line chemotherapy, were included. Patients with disease progression were excluded.

Liver lesions included in the present study were categorized into three groups with reference to the baseline MR examination [18]:DLM in the case of vanished hepatic metastasis;Residual tiny liver metastases (RTLM) in the case of residual hepatic metastases ≤5 mm;Small residual liver metastases (SRLM) in the case of residual hepatic metastases >5 mm and ≤10 mm.

Hepatic lesions larger than 10 mm were excluded from analysis.

### 2.2. MR Imaging Protocol

MR examinations were performed using a 3.0 T superconductive scanner (GE Discovery MR750; GE Healthcare, Milwaukee, WI, USA). An 8-channel phased-array body coil was used for both excitation and signal reception. Our standard MR imaging liver protocol first included axial T2-weighted and in-/out-of-phase axial T1-weighted sequences. Axial DW-MRI was acquired through the entire liver using a single-shot spin-echo echo-planar (SE-EPI) sequence with multiple *b* values (0, 150, 500, 1000, 1500 s/mm^2^), parallel imaging technique and with diffusion-weighted gradients applied in all three orthogonal directions. Subsequently, a 3D breath-hold fat-suppressed T1-weighted LAVA (Liver Acquisition with Volume Acceleration) flex sequence was obtained before and after Gd-EOB-DTPA (Primovist^®^, Bayer HealthCare, Berlin, Germany) injection, including both dynamic and hepatobiliary phases. Primovist was administered intravenously at a rate of 2 mL/s for a total dose of 0.1 mL/kg body weight, followed by a 25 mL saline washout.

### 2.3. Image Analysis

Lesion size was defined as the maximum diameter of the metastasis on the hepatobiliary phase imaging. Low signal intensity on T1-weighted image, moderate signal intensity on T2-weighted image, high signal intensity on DW-MRI and low signal intensity both on dynamic and hepatobiliary contrast-enhanced MRI were used as imaging criteria for detection of CRLM. In addition, sinusoidal obstruction syndrome (SOS), liver damage caused by treatment with oxaliplatin, was detected due to the presence of reticular hypointensity of liver parenchyma in the hepatobiliary phase. Imaging was evaluated by consensus reading of three radiologists (two with 5 and one with more than 20 years of experience in abdominal MRI), blinded with regard to clinical information and outcome.

The outcome of resected liver metastases was assessed in terms of pathological response, while lesions left in situ were evaluated in terms of local relapse or progression according to RECIST criteria version 1.1. Tumor reassessments were performed every 8 weeks by means of CT scan; liver MRI was utilized as problem-solving diagnostic method.

### 2.4. Statistical Analysis

Descriptive statistics were used to summarize the pathological response of resected metastases. The cumulative relapse or progression rates at 6 and 12 months for non-resected DLM and residual metastases, respectively, were estimated using the Kaplan–Meier method, starting from the date of liver MRI performed during first-line chemotherapy until relapse or progression of each lesion. Liver lesions not relapsed or progressed at last follow-up visit or at the beginning of a subsequent line of chemotherapy (i.e., in case of disease progression of patient without relapse or progression of the specific liver lesion) were censored at the date of the last radiological assessment. Log-rank test was used to compare the cumulative relapse rate based on the presence of SOS in DLM group. A two-sided p-value less than 0.05 was considered statistically significant. All statistical analyses were carried out using MedCalc version 14.8.1 (MedCalc Software Ltd., Ostend, Belgium).

## 3. Results

Among 120 patients with CRLM undergoing liver MRI, 64 patients performed this exam during first-line chemotherapy. Fifty-two patients had at least a DLM or a residual metastasis less than or equal to 10 mm and were included in our analysis (Figure 1).

Clinical characteristics of the patient population are summarized in Table 1. Most patients had synchronous disease (73%), with a left or rectum primary tumor (83%).

First-line chemotherapy regimens used were mostly triplet plus anti-VEGF or anti-EGFR in 16 patients (31%) and doublets plus anti-EGFR in 15 patients (29%), while 13 (25%) and 8 (15%) patients were treated with doublets plus anti-VEGF and doublets alone, respectively.

Out of 265 liver lesions radiologically reviewed, 185 metastases fulfilled the inclusion criteria of this study: 40 DLM, 82 RTLM and 60 SRLM, as reported in Figure 1.

### 3.1. DLM Results

Overall, 15 patients had at least a DLM. The median number of DLM was 1.5 (IQR: 1–3). The median time interval between diagnosis of liver lesions at baseline MRI and the detection of DLM at MRI was 6.6 months (IQR: 4–8.7 months). Among 40 DLMs, the site where DLM were located was resected in four cases (two in one patient and another two in two different patients). A complete pathological response was observed in three (75%) instances (Table 2).

For the other 36 DLMs, at a median follow-up period of 8.8 months, local relapse occurred in 12 cases (33%), with 6- and 12-month cumulative relapse rates of 26% and 43%, respectively (Figure 2, panel A). Out of 36 DLMs left in situ, 9 (25%) lesions were associated with SOS at liver MRI, with local relapse occurring in 6 cases (67%). A higher 6-month cumulative relapse rate was observed in cases with SOS compared to those without SOS (67% vs. 11%, HR: 11.2; 95%CI: 2.32–54.20, *p* = 0.003) (Figure 2, panel B).

### 3.2. RTLM Results

Overall, 33 patients had at least an RTLM (Figure 3). The median number of RTLM was 2 (IQR: 1–3.2). The median time interval between diagnosis of liver lesions at baseline MRI and the detection of RTLM at MRI was 5.3 months (IQR: 4.1–7.5 months). Out of 82 RTLMs, 48 were surgically removed with a pCR rate of 44% (Table 2). Among 34 RTLMs left in situ, at a median follow-up period of 11.2 months, local relapse occurred in 10 lesions (29%), with 6- and 12-month cumulative relapse rates of 19% and 37%, respectively (Figure 4, panel A).

### 3.3. SRLM Results

Overall, 29 patients had at least an SRLM (Figure 3 and Figure 5). The median number of SRLM was 2 (IQR: 1–2.2). The median time interval between diagnosis of liver lesions at baseline MRI and the detection of SRLM at MRI was 5.5 months (IQR: 4.2–7.2 months). Out of 60 SRLMs, 37 were surgically removed with a pCR rate of 38% (Table 2). Among 23 SRLMs left in situ, at a median follow-up period of 16.4 months, local relapse occurred in 13 lesions (57%), with 6- and 12-month cumulative relapse rates of 45% and 58%, respectively (Figure 4, panel B).

## 4. Discussion

For decades, CRLM were listed as a non-curable medical finding. Surgical techniques for liver surgery, systemic treatment regimens and perioperative care have significantly improved over time. Systemic chemotherapy and targeted therapy have significantly increased the chance of conversion to surgery for patients with initially unresectable colorectal liver metastases, resulting in improved survival. 

The management of DLM and small remnant of liver metastases after chemotherapy has always been a matter of debate [19,20]. Recently, this problem has become increasingly common due to the current availability of highly active pre-operative chemotherapy regimens in combination with targeted agents, resulting in a dimensional dramatic response and in the disappearance of several liver metastases in a non-negligible group of patients [20]. In addition, the introduction of new liver surgical techniques, such as organ-sparing resection, increased the number of patients eligible for liver metastasectomy as well as questioning the surgical management of DLM and small remnants of liver metastases, which could not be detected intraoperatively. In these cases, an aggressive surgical approach attempting the resection of all the sites where DLM and small remnants of liver metastases were located could reduce the liver remnant health and the possibility of further liver surgery [21,22,23,24,25].

Several retrospective studies addressed this question [10]. Although it seems quite clear that radiological complete response assessed by CT scan corresponds to a real complete response only in a minority of cases, this correlation may increase using liver MRI [26]. Recent clinically important advances in liver MRI include the addition of diffusion-weighted imaging and hepatobiliary contrast agents such as gadoxetic acid. In a recent meta-analysis of the literature, in which thirty-nine articles were included, Vilgrain et al. [27] concluded that in patients with CRLM, combined DW-MRI and gadoxetic acid-enhanced MR imaging has the highest sensitivity for detecting liver metastases on a per-lesion basis; the same results are obtained in liver metastases smaller than 1 cm. In this regard, studies specifically comparing CT scan and liver-specific contrast-enhanced MRI showed higher correspondence between DLM in imaging and pCR or absent from in situ recurrence in favor of MRI (65–85% versus 35–59%) [28,29]. Our results, albeit limited by a small sample size, are in line with literature data, with a pCR rate of 75% in resected DLM and with 33% of local relapse for DLM left in situ. In addition, our results showed that the in situ recurrence decreases in the case of an absence of SOS [30]. Indeed, the presence of reticular hypointensity, a feature associated with chemotherapy-induced SOS, may reduce the accuracy of liver-specific contrast-enhanced MRI for evaluation of CRLM [31]. On the other hand, SOS could be directly responsible for the relapse with a still unknown mechanism. These data were confirmed by the study of Kim et al. [10], where the in situ relapse decreased from 15% to 5% in the case of homogenous signal intensity of liver parenchyma on hepatobiliary phase images, while it increased from 15% to 36% in the case of reticular hypointensity.

Regarding the small remnants of liver metastases after chemotherapy, our results showed a risk of relapse of 29% and 57% for RTLM and SRLM left in situ, respectively, and a pCR rate of about 40% overall for resected lesions [32]. Similar data regarding RTLM were reported by Kim et al. that showed a pCR rate of 41% for resected lesions and a risk of relapse of 31% for metastases left in situ.

In our series, all MR exams were performed with a state-of-the-art magnet operating at 3.0 T, and a multiparametric MR imaging protocol, including both diffusion-weighted imaging and hepatobiliary phase, was utilized. In this way, very small liver metastases could also be identified. However, MRI is very susceptible to breathing artefacts, especially in the segments of the hepatic dome; in the case of significant artifacts, the evaluation of liver parenchyma may be not reliable, and the detection of small tumor foci can be extremely difficult. As suggested by Beyer et al. [33], in these selected cases, CEUS could improve diagnostic accuracy since a combination of both imaging modalities might be able to help reduce the number of false-negative results; CEUS could also be helpful in the detection of intralesional cystic necrosis. The role of FDG PET-CT when added to cross-sectional imaging to assess response in CRLM is undetermined taking into account the well-known low sensitivity for lesions < 1 cm; on the other hand, it may be a problem-solving technique in detecting extrahepatic disease in pre-surgical restaging.

Our study has several limitations, including the retrospective nature, the low sample size, the limited follow-up and the use of CT scan as follow-up imaging for non-resected lesions that could have underestimated the risk of relapse.

Although no definitive conclusions can be drawn, our results, in addition to the literature data, can provide same suggestions for the proper management of DLM and small remnants of liver metastases after chemotherapy. In particular, since DLM assessed through hepatobiliary contrast-enhanced and diffusion-weighted MRI indicates a real complete response with high probability in patients without chemotherapy-induced SOS, follow-up without resection can be considered for these cases [34,35]. Indeed, no difference in terms of survival was observed in a study comparing patients with resected DLM versus patients with DLM left in situ, as assessed by pre-operative MRI [34,35]. This may also be due to the possibility of subsequent liver surgery in the case of local recurrence [36,37,38]. Conversely, considering the high risk of relapse and the low rate of pCR for RTLM and SRLM, maximum effort should be made for resection of small remnants of liver metastases.

## 5. Conclusions

The treatment plan for patients with CRLM should be determined case by case in a multidisciplinary setting. Adequate radiological staging and restaging after chemotherapy are crucial for the optimal selection of patients as candidates for surgery and to individualize the optimal treatment strategy. DLM assessed via hepatobiliary contrast-enhanced and diffusion-weighted MRI very probably indicates a complete response in patients without chemotherapy-induced SOS. In these patients, a follow-up with liver MRI can be considered, and resection should be performed in the case of disease relapse. The surgical removal of small remnants of liver metastases (less than or equal to 10 mm) should always be advocated whenever technically possible.

## Figures and Tables

**Figure 1 cancers-15-02200-f001:**
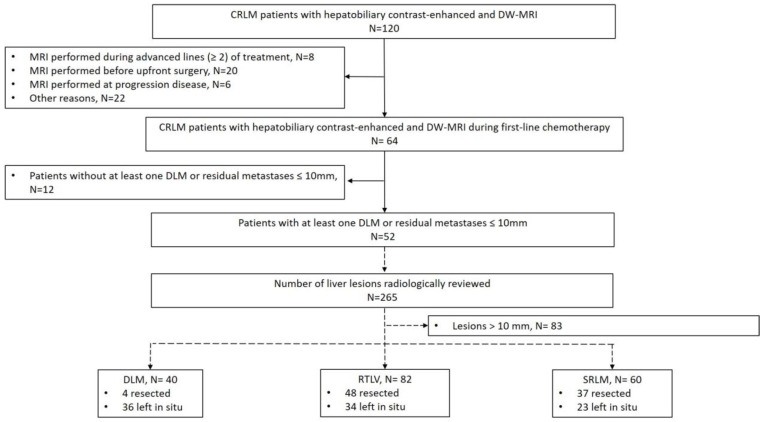
CRLM patients’ flowchart.

**Figure 2 cancers-15-02200-f002:**
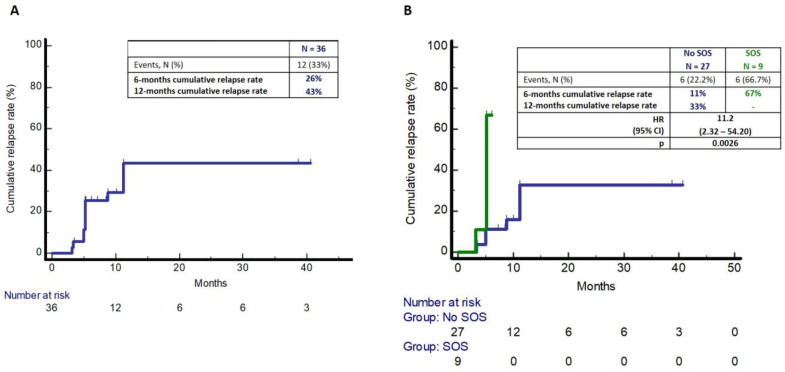
Cumulative relapse rate for DLM (**panel A**) and according to SOS (**panel B**).

**Figure 3 cancers-15-02200-f003:**
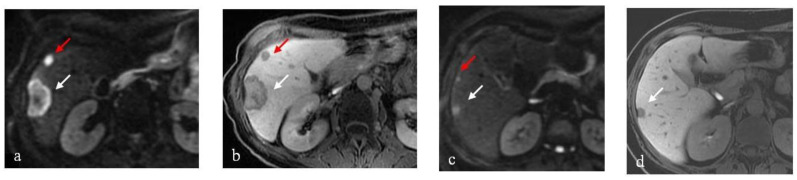
A 53-year-old male who had an RTLM and SRLM after chemotherapy. Two liver metastases in segment V sized 32 mm and 12 mm (white and red arrows, respectively) were identified on diffusion-weighted (**a**) and hepatobiliary phase (**b**) images of baseline MRI. After chemotherapy, the size of the bigger metastasis decreased to 9 mm (white arrow), whereas the smaller one became visible (red arrow) only on diffusion-weighted imaging (**c**) but not on hepatobiliary phase image (**d**) of follow-up MRI.

**Figure 4 cancers-15-02200-f004:**
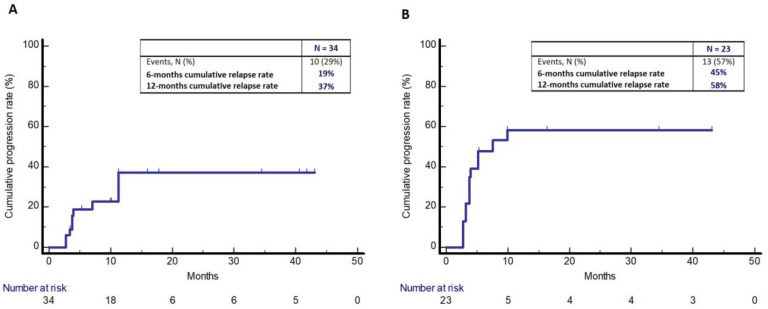
Cumulative progression rate of RTLM (**panel A**) and SRLM (**panel B**).

**Figure 5 cancers-15-02200-f005:**
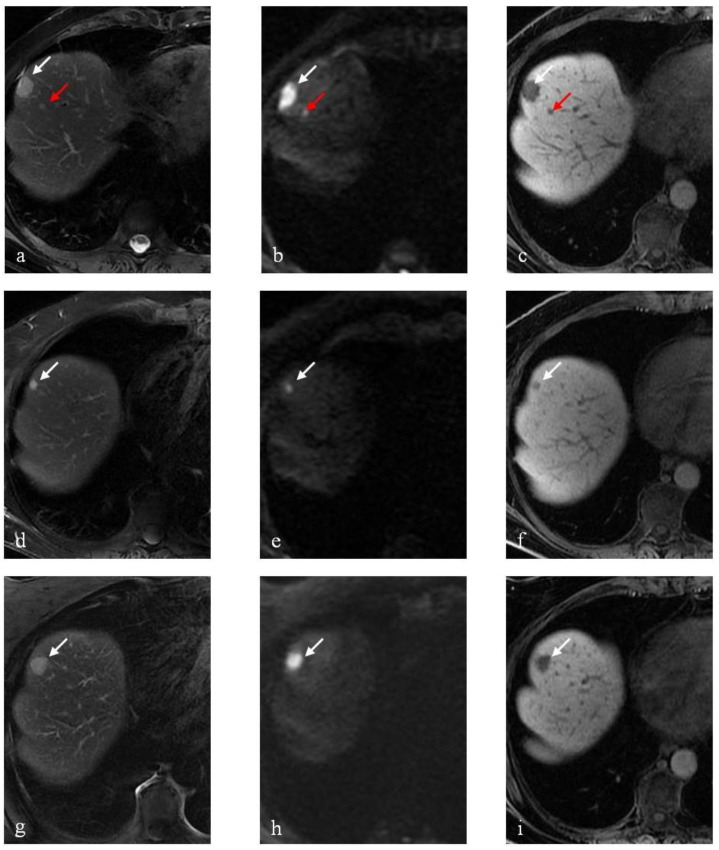
A 45-year-old female who had a DLM and SRLM after chemotherapy. Two liver metastases in segment VIII sized 18 mm and 4 mm (white and red arrows) were observed on T2-weighted (**a**), diffusion-weighted (**b**) and hepatobiliary phase (**c**) images of baseline MRI. After chemotherapy, the size of the bigger metastasis decreased to 7 mm (white arrow), while the smaller one became invisible on T2-weighted (**d**), diffusion-weighted (**e**) and hepatobiliary phase (**f**) images of follow-up MRI. The SRLM showed progression on T2-weighted (**g**), diffusion-weighted (**h**) and hepatobiliary phase (**i**) images of a 2-year follow-up MRI; the DLM did not show any recurrence.

**Table 1 cancers-15-02200-t001:** Patient characteristics.

Characteristics, N (%)	N Tot = 52
Sex	
*Male*	26 (50)
*Female*	26 (50)
**Age, years, median [range]**	59 (25–77)
**ECOG PS**	
*0*	47 (90)
*1–2*	5 (10)
**Median number of liver lesion [range]**	4 (1–17)
**Location of primary tumor**	
*Right colon*	9 (17)
*Left colon or rectum*	43 (83)
**Primary tumor resected**	
*Yes*	30 (58)
*No*	22 (42)
**Prior adjuvant therapy**	
*Yes*	8 (15)
*No*	44 (85)
**Time to metastasis**	
*Synchronous*	38 (73)
*Metachronous*	14 (27)
**Molecular status**	
*RAS/BRAF wild-type*	27 (52)
*RAS mutated*	23 (44)
*BRAF mutated*	1 (2)
*Unknown*	1 (2)
**MSI status**	
*MSS*	45 (86)
*MSI high*	2 (4)
*Unknown*	5 (10)
**Treatment received**	
*Doublets*	8 (15)
*Doublets + antiVEGF*	13 (25)
*Doublets + antiEGFR*	15 (29)
*Triplet + antiVEGF or antiEGFR*	16 (31)

**Table 2 cancers-15-02200-t002:** pCR rate of resected colorectal liver metastases.

Resected Colorectal Liver Metastases N=89			
	DLM N = 4	RTLM N = 48	SRLM N = 37
pCR	3 (75%)	21 (44%)	14 (38%)
No pCR	1 (25%)	27 (56%)	23 (62%)

## Data Availability

Authors agree to make data and materials supporting the results presented in this paper available upon reasonable request.

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
