# Peer review of "Exploring the Outcome of Disappearance or Small Remnants of Colorectal Liver Metastases during First-Line Chemotherapy on Hepatobiliary Contrast-Enhanced and Diffusion-Weighted MR Imaging"

_cancers, 2023, doi:10.3390/cancers15082200_

Round 1

Reviewer 1 Report

The authors described the usefulness of MRI in the evaluation of colorectal cancer liver metastases that was disappeared or nearly disappeared. CT alone was insufficient to evaluate micrometastases and contrast-enhanced MRI and diffusion-weighted MRI could improve diagnostic accuracy. They classified micrometastases smaller than 10 mm into DLM, RTLM, and SRLM, and described the pCR rate for each and the recurrence rate if not resected.

Although not a large study, this study is highly reliable because each case was examined in great detail. The results of this study are extremely useful in deciding whether or not to perform resection of small lesions after chemotherapy.

Although this is a retrospective study and the number of cases is small, the authors also discussed the limitation of the study in the discussion, which is considered sufficient for the article.

Reviewer 2 Report

MRI can be very susceptible to breathing, especially in Segmnet VIII. This then leads to significant artifacts. This makes the detection of small tumor foci more difficult. CEUS can be helpful for this purpose. There is a publication on this by Beyer LP et al in Ultrasound in Medicine.

The detection of small tumor foci depends on field strength, coil technique and calculation of the K-space. This has to be discussed.

Diffusion is difficult when the liver is cirrhotic and ascites is present. Moreover, the calculation of B-values and ADC-image in different devices must be adjusted first.

Liver specific contrast agent can better differentiate lesions that are complicated cystic or partial necrotic. This should be discussed.

Foci up to 10 mm in HCC are difficult to detect by diffusion alone.

Diffusion MRI is not widely available. Therapy monitoring should also correlate with ultrasound and CEUS. 

A complete response could only be proven by histology or a sufficiently long follw up without image morphologic recurrence. The importance of therapy monitoring also by PET-CT should be discussed. 

Lesions with cystic  necrosis may complicate imaging, if possible.
